# Comparative Analysis of Codon Usage Patterns in Nuclear and Chloroplast Genome of *Dalbergia* (Fabaceae)

**DOI:** 10.3390/genes14051110

**Published:** 2023-05-19

**Authors:** Zu-Kai Wang, Yi Liu, Hao-Yue Zheng, Min-Qiang Tang, Shang-Qian Xie

**Affiliations:** Key Laboratory of Genetics and Germplasm Innovation of Tropical Special Forest Trees and Ornamental Plants (Ministry of Education), Hainan Key Laboratory for Biology of Tropical Ornamental Plant Germplasm, School of Forestry, Hainan University, Haikou 570228, China

**Keywords:** *Dalbergia*, genome, CUB, gene expression, phylogenetics

## Abstract

The *Dalbergia* plants are widely distributed across more than 130 tropical and subtropical countries and have significant economic and medicinal value. Codon usage bias (CUB) is a critical feature for studying gene function and evolution, which can provide a better understanding of biological gene regulation. In this study, we comprehensively analyzed the CUB patterns of the nuclear genome, chloroplast genome, and gene expression, as well as systematic evolution of *Dalbergia* species. Our results showed that the synonymous and optimal codons in the coding regions of both nuclear and chloroplast genome of *Dalbergia* preferred ending with A/U at the third codon base. Natural selection was the primary factor affecting the CUB features. Furthermore, in highly expressed genes of *Dalbergia odorifera*, we found that genes with stronger CUB exhibited higher expression levels, and these highly expressed genes tended to favor the use of G/C-ending codons. In addition, the branching patterns of the protein-coding sequences and the chloroplast genome sequences were very similar in the systematic tree, and different with the cluster from the CUB of the chloroplast genome. This study highlights the CUB patterns and features of *Dalbergia* species in different genomes, explores the correlation between CUB preferences and gene expression, and further investigates the systematic evolution of *Dalbergia*, providing new insights into codon biology and the evolution of *Dalbergia* plants.

## 1. Introduction

The central dogma is the fundamental principle of the transfer of genetic information between macromolecules within biological cells [1]. During the transfer of genetic information from the mRNA to the protein, the triplet codons play a crucial role in the formation of proteins in organisms, especially during the translation process [2]. Three bases make up a codon and jointly encode one amino acid, with each amino acid being encoded by one or more codons, but no more than six [3]. Tryptophan (Trp, W) is only encoded by UGG, and methionine (Met, M) is only encoded by AUG. The remaining 18 amino acids are encoded by multiple synonymous codons, ensuring the stability of the translation process [4]. However, synonymous codons are not used randomly or equally; rather, some are preferentially used to encode amino acids over others. This phenomenon of using synonymous codons with different frequencies is known as codon usage bias (CUB) [5,6,7,8]. The preference for certain codons mainly reflects the impact of translation levels under weak natural selection. Furthermore, the composition of the third base of a codon is subject to strong mutational bias and codirectional natural selection [9,10].

The chloroplast (cp) genome is non-recombinant and maternally inherited, exhibiting features such as semi-autonomy and conservative gene content, organization, and structure relative to the mitochondrial and nuclear genomes [11,12]. It is relatively stable in structure and contains a large amount of genetic information specific to plants. Therefore, it plays an important role when we study the origin, systematic evolution, species identification, and classification of plants and serves as a critical data source for exploring the evolutionary relationships between plant species [13,14]. Systematic phylogenetic analysis based on chloroplast genomes has been extensively reported, for instance, *Achyranthes* (Amaranthaceae) [15], *Ferula* (Apiaceae) [16], and *Aquilegia* (Ranunculaceae) [17]. Plants are subject to different evolutionary constraints, and there are differences in codon usage patterns in chloroplast genomes [18].

*Dalbergia* is a pan-tropical genus with over 269 recognized tree, shrub, and woody vine species [19]. The genus is native to more than 130 countries in tropical and subtropical regions [20]. Members of *Dalbergia* have high economic and medicinal values [19]. For example, the rare species *D. odorifera* [20,21] and the high-quality timber species *Dalbergia cochinchinensis* [22]. Studying codon usage bias in *Dalbergia* may have practical applications for future genetic modification of these species, potentially as timber, furniture, or other economically valuable products [23,24]. Furthermore, *Dalbergia* plants possess nitrogen-fixing capabilities and can serve as support species to produce parasitic trees such as sandalwood [25]. CUB analysis is an effective method for studying species specificity, evolutionary relationships, and mRNA translation, and for discovering new genes [26]. Therefore, we systematically analyzed the codon usage patterns of the entire genome of *D. odorifera* and the protein-coding sequences of 25 chloroplast genomes of the *Dalbergia* genus, and evaluated the impact of natural selection, mutational pressure, and other factors on codon usage. At the same time, we investigated the relationship between codon usage preference in the entire genome of *D. odorifera* and gene expression, as well as the systematic evolution of the *Dalbergia* genus. This study aims to provide valuable insights into the genetic modification and improvement of *Dalbergia* species for economic and environmental purposes and to contribute to a better understanding of the codon usage bias in other plants.

## 2. Materials and Methods

### 2.1. Data Collection and Processing

The protein coding sequences (CDS) of the chloroplast reference genome of *Dalbergia* were collected and downloaded by us from the NCBI database (Appendix A). The nuclear genome protein coding sequence of *D. odorifera*, and the RPKM expression matrix of four tissues (root, stem, leaf, and seed) were downloaded directly from the GigaDB dataset (http://gigadb.org/dataset/100760), accessed on 25 May 2022. To minimize errors caused by short sequences, CDS sequences less than 300 bp in length were removed [27].

### 2.2. Codon Usage Indicators

Codonw1.4.2 and the online software Cusp were used to calculate codon usage metrics [28,29]: ENC, RSCU, GC, GC3s, A3, T3, G3, C3, GC1, GC2 and GC3. The GC content of different CDS sequence positions were calculated by a Python3.6 script.

### 2.3. Synonymous Codon Analysis

The relative synonymous codon usage (*RSCU*) is the ratio of the observed frequency of a particular codon to the expected frequency if all synonymous codons for a specific amino acid were used equally in the coding sequence. A codon with an *RSCU* value > 1.6 is considered over-represented, while an *RSCU* value < 0.6 is considered under-represented. *RSCU* values < 1 indicate that the codon is used less frequently, while *RSCU* values >1 indicate that the codon is used more frequently. The *RSCU* index of codons is calculated as follows [30]:$${RSCU}_{ij}=\frac{X_{ij}}{\frac{1}{n_{i}}\sum_{j=1}^{n_{i}}X_{ij}}.$$

In which *X_ij_* is the frequency of occurrence of the *j*th codon of the *i*th amino acid and *n_i_* is the number of codons of the *i*th amino acid.

### 2.4. ENC-Plot Analysis

The effective codon number (*ENC*) is usually used to express the degree of random selection of codon usage bias. It takes values in the range of 20–61. The magnitude of *ENC* value is inversely proportional to the degree of codon usage bias. That is, a smaller *ENC* value is associated with a strong codon bias, and conversely, a larger *ENC* value is associated with a weak codon usage bias. We used the values of *GC*3 and *ENC* as the horizontal and vertical coordinates, respectively, and added the standard curve by the following equation [31]:$${ENC}_{\exp}=2+GC3s+\frac{29}{{GC3s}^{2}+{\left(1-GC3s\right)}^{2}}$$

*ENC*_exp_ represents the expected position of the gene when codon use preference is determined only by *GC*3s composition. *GC*3s represents the content of the third base *G* + *C* of the synonymous codon.

### 2.5. Parity Rule 2 (PR2) Bias Plot Analysis

The formation of codon usage preferences is closely related to the bases in the third position of the codon. We calculated the base composition at the third position of the codon. We also used G3/(G3 + C3) as the horizontal coordinate and A3/(A3 + T3) as the vertical coordinate, thus analyzing the distribution of the third base in the codon. Theoretically, the frequency of using the third base of the codon should follow the PR2 principle (A = T, C = G). Additionally, the bases that deviate from the central distance indicate the degree and direction of codon usage deviation from the rule [8].

### 2.6. Neutrality Plot Analysis

Neutral plot analysis is a method used to explore factors that influence codon usage patterns. This method involves calculating GC3 and GC12 (the mean of GC1 and GC2), with GC3 as the independent variable and GC12 as the dependent variable, and fitting a straight line to the data. The regression coefficient is a key indicator of neutrality, with a positive or negative value indicating the direction of the correlation between GC3 and GC12, and its magnitude determining the strength of the correlation [32].

### 2.7. The Determination of Optimal Codon

Optimal codons are those with high frequency of use and an ENC difference above a certain threshold. High utilization codons are those with RSCU values greater than 1, while the ENC difference categorizes genes into high and low codon preference groups based on their ENC values. We also calculated the ΔRSCU value of codons, with a threshold value of 0.08. If a codon has ΔRSCU > 0.08 and RSCU > 1, it is considered an optimal codon [33].

### 2.8. The Correlation between Codon Usage and Gene Expression

In order to investigate the interaction between codon usage and gene expression in the nuclear genome of *D. odorifera*, we evaluated three levels: (1) At the sequence level, all ENC values of the CDS sequence were divided into two categories (low and high codon bias strength). They were also characterized at high, medium, and low transcription levels, respectively. (2) At the codon level, the bias of bases (A, T, C, and G) in the third position of synonymous codons was calculated. The total number of four bases in each codon and the maximum number of biases in the third position of the codon were calculated, and then all RPKMs of CDS sequences were classified into four categories based on the use of the third base of the codon. (3) At the amino acid level, amino acids were divided into four categories based on the frequency of use of 59 synonymous codons (Appendix A). For each type of amino acid, the RSCU value of synonymous codons was calculated, and the codon with the highest RSCU was used to group them. The gene expression level of CDS sequences in each group was then analyzed, and t-tests were used to analyze the significant differences in the usage of different categories of synonymous codons for each type of amino acid [34].

### 2.9. Cluster Analysis and Phylogenetic Tree Construction

Clustering analysis was performed based on codons using RSCU values as features with the Euclidean distance method [35]. In addition, 25 chloroplast genome sequences were used for phylogenetic analysis. All sequences were aligned using MAFFT (v7.505) (https://mafft.cbrc.jp/alignment/server/, accessed on 25 May 2022) with the parameters “mafft—thread 8—threadtb 5—threadit 0—reorder—auto”. The phylogenetic tree was constructed using MEGA v11.0 (https://www.megasoftware.net/, accessed on 25 May 2022), where obvious regions that may not belong to the pair were manually removed using the neighbor-joining (NJ) method [36].

## 3. Results

### 3.1. Analysis of Codon Composition Characteristics

In the nuclear genome of *D. odorifera*, there are 27,940 coding protein sequences, while the 25 chloroplast genomes of the *Dalbergia* species have a total of 1438 protein-coding sequences (Appendix A). For each sequence, the usage of seven codons was calculated, and the codon usage biases of the protein-coding sequences in the nuclear genome and chloroplast genomes were compared (Table 1 and Figure 1). Additionally, a correlation analysis was conducted on the codon usage bias of protein-coding sequences in both the nuclear genome and chloroplast genomes (Figure 2A,B).

In the nuclear genome sequence, there were 86 sequences with an ENC value less than 35, and 19,724 sequences with an ENC value greater than 50. The average ENC value was 51.86 (Table 1 and Appendix A), indicating a weak overall codon usage bias. In the protein-coding sequences of the chloroplast genome, there were no sequences with an effective number of codons (ENC) value less than 35, while 684 sequences had an ENC value greater than 50. The average ENC value was 49.90, which was slightly lower than that observed in the protein-coding sequences of the nuclear genome (Table 1 and Appendix A). The average GC content of the nuclear genome protein-coding sequences was 46.1%, while that of the chloroplast genome protein-coding sequences was 39.0% (Table 1). The total GC content of the chloroplast genome protein-coding sequences was lower than that of the nuclear genome sequence (t = 58.672, *p* < 0.0001). Furthermore, differences were found in the GC content relationship at different positions within the protein-coding sequences of both the genome-wide and chloroplast genomic sequences. Interestingly, while the GC1 content was highest in the chloroplast genome and the GC3 content was lowest, the opposite was observed in the genome-wide sequences, with GC1 being highest and GC2 being lowest. These results are consistent with previous studies on *Mesona chinensis* [37] and *Sesamum indicum* [38] (Figure 2C,D).

In addition, both the nuclear genome and the chloroplast genome protein-coding sequences had GC content and GC3s less than 50%, indicating an enrichment of A/U bases in the third position of the codon. However, there was a significant difference in GC3s between the chloroplast and nuclear genome protein-coding sequences (t = 26.726, *p* < 0.001), indicating differences in codon usage between the two sequence types. Meanwhile, our correlation analysis shows that the chloroplast genome protein-coding sequences showed a stronger correlation in codon usage bias compared to the nuclear genome protein-coding sequences, particularly in the GC1, GC2, and ENC indices. Notably, the ENC value of both showed a positive correlation with the G/C content and a negative correlation with the A/T content. Additionally, the ENC value was most closely related to GC3 content (Figure 2A,B), indicating a strong conservation of the GC3 at the genetic level.

Furthermore, the analysis of codon composition characteristics in the chloroplast genome of 25 *Dalbergia* species revealed an average of 24,482 codons, with *Dalbergia millettii* having the most (24,738) and *Dalbergia oliveri* having the fewest (24,317) (Appendix A). The GC content of the three base positions in each chloroplast genome’s codons was biased toward A and U. The ENC values of all 25 species were between 49 and 50 (Appendix A), indicating a weak codon bias. These findings suggest that the codon GC content, ENC value, and their correlations may affect the analysis of factors influencing codon usage.

### 3.2. Synonymous Codon Analysis

We analyzed synonymous codon usage in 25 species of *Dalbergia* and found differences in their relative usage across species, which allowed us to group them into nine clusters. The 59-dimensional codon vectors could be classified into six categories. In the chloroplast genomes of *Dalbergia*, we observed a bias toward using high-frequency codons with A/U endings, with 30 synonymous codons having a relative usage greater than 1 (Figure 3A).

### 3.3. ENC—Plot Analysis

Mutational pressure and natural selection are important factors that affect codon usage. If mutational pressure is the main factor affecting codon usage preference, the true ENC values (ENCobs) should be close to the region of the ENCexp expectation curve. Conversely, if natural selection is the main factor affecting codon usage preference, the true ENC values will deviate farther from the region of the ENCexp expectation curve [39]. As shown in the figure, Both the protein-coding sequences of the nuclear genome and the chloroplast genome are mainly distributed below the standard curve, and the ENC values are clustered between 40 and 61. This indicates that both have similar codon usage preferences and relatively weak codon bias. In addition, natural selection is the main factor affecting their ENC usage preferences, followed by mutation (Figure 3B).

### 3.4. PR2-Plot Analysis

This study conducted a Parity Rule 2 (PR2) analysis of the relationship between the third base of codons (A3/T3 and G3/C3) in the protein-coding sequences of the nuclear genome of *D. odorifera* and 25 chloroplast genomes of the *Dalbergia*. The results showed that the coordinate points of the four regions were unevenly distributed. Specifically, in the protein-coding sequences of the nuclear genome of A. sinensis, the A3/(A3 + T3) ratio was mainly distributed below 0.5, indicating that the frequency of T base usage was higher than that of A in the nuclear genome. In addition, in the chloroplast genomes of the *Dalbergia* species, the G3/(G3 + C3) ratio was mainly distributed above 0.5, indicating that the frequency of G base usage was higher than that of C in the chloroplast genome (Figure 3C). These results further demonstrate that base mutations and natural selection jointly affect the codon usage bias in both the nuclear genome and chloroplast genome of the *Dalbergia* species, with natural selection being the major factor influencing the codon usage bias of these genomes.

### 3.5. Neutrality Plot Analysis

In neutral plot analysis, if the main factor affecting codon bias is mutation, then the fitted linear regression coefficient is close to 1. Conversely, if the main factor affecting codon bias is natural selection, then the fitted linear regression coefficient should be close to 0 [40]. Neutral plot analysis showed that in the protein-coding sequences of the nuclear genome (r = 0.2864, *p* < 0.001), GC12 was distributed from 0.2610 to 0.8219, and GC3 was distributed from 0.1877 to 0.9112. In the protein-coding sequences of the chloroplast genome (r = 0.2308, *p* < 0.001), GC12 was distributed from 0.3046 to 0.5396, and GC3 was distributed from 0.2005 to 0.4805 (Appendix A). The fluctuations in GC3 and GC12 are smaller in the chloroplast genome and their values tend to be more stable. The correlation coefficients of GC3 and GC12 in the protein-coding sequences of the nuclear genome and chloroplast genome were 0.1294 and 0.3118, respectively (Figure 3D), indicating that both parameters were positively correlated in the protein-coding sequences of the nuclear genome and chloroplast genome, with a closer correlation in the chloroplast genome. The results further indicate that natural selection is the major factor influencing the codon usage patterns in both the nuclear and chloroplast genomes of the Dalbergia genus, while base mutations play a secondary role.

### 3.6. Optimal Codon Determination

To determine the optimal codons for the protein-coding sequences of the nuclear genome and the chloroplast genomes of 25 *Dalbergia* species, we selected genes with 5% of each end of the ENC value as the high and low expression genomes, respectively, which were sorted according to the ENC value. The RSCU values of all synonymous codons in the nuclear genome and chloroplast genome were calculated. A total of 25 optimal codons (RSCU > 1, ΔRSCU > 0.08, and 16 ending in A/U (A:6, U:5, G:2, and C:3) was selected from the nuclear genome protein-coding sequences (Appendix A). In the chloroplast genomic protein-coding sequence, there are between 15 and 23 optimal codons for two species, *Dalbergia martinii* and *Dalbergia obtusifolia*, respectively (Figure 4). In addition, the chloroplast genomes of 25 species share four codons: UUG (leucine), UCU (serine), GCA (alanine), and UGU (cysteine). Except for UUG, the third bases of the other three codons preferentially use A/U endings (Figure 4). This result is like the genome-wide protein-coding sequence. These findings suggest that the whole-genome and chloroplast genomic protein-coding sequences of *Dalbergia* tend to use A/U-terminal codons.

### 3.7. Relationship between Codon Usage and Gene Expression

To better understand the expression of genes at the level of codon bias, we plotted RPKM-ENC heatmaps for the leaf, root, stem, and seed tissues of *D. odorifera* (Figure 5). From the figure, we can see that the expression trends of genes in the four tissues are similar, and most genes have ENC values greater than 50. This indicates that the codon bias of the protein-coding sequences in the nuclear genome of *D. odorifera* is relatively weak. To further analyze the relationship between gene expression and codon bias, we conducted a three-level analysis: At the sequence level, we classified all genes into three expression levels: high (RPKM > 10), medium (10 ≥ RPKM > 1), and low (RPKM ≤ 1), with genes having an RPKM of 0 being excluded. Each expression level was then further divided into two groups based on their ENC values. Our results showed that more genes had ENC values greater than 50, indicating a lower codon usage bias in the *D. odorifera* genome. Notably, significant differences in RPKM values were observed in both the high and medium expression level groups (Appendix A). It is worth mentioning that, within our three expression levels, gene expression and codon bias were inconsistent: ① In the high expression level group, gene expression increased with stronger codon bias (t = 3.5113, *p* = 4.519 × 10^−4^; ② In contrast, in the medium expression level group, genes with weaker codon bias had higher expression levels (t = −5.5917, *p* = 2.348 × 10^−8^); ③ In the low expression level group, the relationship between gene expression and codon bias was not evident, unlike in the other two groups (t = −0.9879, *p* = 0.3233). This result suggests that in *D. odorifera*, the stronger the codon bias in highly expressed genes was, the higher was the gene expression level, while such a trend was not observed in the medium and low expression level genes (Figure 6A–C).

Based on the third base bias of each codon, we grouped all CDS sequences into four groups at the codon level: those ending in A, T, C, and G. CDS sequences with codons biased toward G/C endings had higher RPKM values than those biased toward A/T endings (Figure 6D and Appendix A). This result suggests that genes biased toward G/C endings at the third base tend to be highly expressed.

On the basis of the frequency of synonymous codon usage for each amino acid, we divided the CDS sequences into two-codon groups (Phe, His, Lys, Asn, Asp, Cys, Gln, Glu, and Tyr) (Appendix A), three-codon groups (Ile), four-codon groups (Pro, Thr, Ala, Gly, and Val), and six-codon groups (Ser, Leu, and Arg) (Appendix A and Appendix A). For each compared synonymous codon of an amino acid, except for Leu and Arg, codons ending in G/C were predominantly expressed compared to those ending in A/U (Appendix A). This result is consistent with the codon-level findings. It is worth noting that although the synonymous codon analysis and optimal codon analysis suggest that *D. odorifera* prefers to end with A/U, most highly expressed genes in this study tended to end with G/C. This suggests that highly expressed genes in *D. odorifera* prefer to end with G/C, unlike the codon third base bias preference for A/U.

### 3.8. Phylogenetic Relationships of 25 Dalbergia Species

To investigate the correlation between codon usage bias and phylogeny, we generated phylogenetic trees using RSCU, CDS, and whole chloroplast genomes. Our analysis revealed that the phylogenetic tree constructed based on the chloroplast nuclear genomes and CDS sequences of the 25 *Dalbergia* species was more consistent with the true classification of the species, indicating a closer relationship between codon usage preference and evolutionary history. These findings could provide insights into the evolution and diversification of the *Dalbergia* (Figure 7).

The clustering results based on the RSCU values of codon preference features showed significant differences, with only some branches being identical (Figure 7 and Appendix A). However, the preferences for GC content and ENC usage were consistent regardless of the evolutionary tree, indicating that each species in Dalbergia has a fixed codon usage profile. Although GC3 exhibits slight fluctuations and is more like the CDS sequence based on the whole chloroplast genome, both differ from the clustering tree constructed based on RSCU. This suggests that preference-free codons also play a significant role in the process of species evolution, and the relevant properties of preference-free codons need further investigation (Figure 7 and Appendix A).

## 4. Discussion

Unequal use of synonymous codons in coding sequences is common in all life forms [41]. According to previous studies, codons play an important role in gene regulation and molecular evolution as important constituent elements in the translation of gene coding regions into shape proteins. The coupling between codon usage and protein sequence selection varies over a wide range and is particularly pronounced in high preference genes [42,43]. At present, studies of codon usage bias have been reported for many species [39,44,45,46]. In this study, we analyzed the codon usage patterns in both the nuclear genome and chloroplast genome of *D. odorifera* and 25 other species of the *Dalbergia*. We found that the codon usage patterns were very similar among the nuclear genomes and chloroplast genomes of the *Dalbergia*, as well as among different chloroplast genomes within the genus. Furthermore, we evaluated the influence of factors such as mutation pressure and natural selection on the uneven usage of codons. We also conducted an in-depth exploration of the correlation between codon usage bias in the nuclear genome of A. sinensis and its gene expression. Additionally, we constructed phylogenetic trees based on RSCU, CDS, and the whole chloroplast genome, respectively.

Due to the relatively weak selection pressure on the third base of codons, GC3 is usually considered an important parameter for analyzing codon usage bias [47]. Based on our analysis of synonymous and optimal codons, we discovered that the protein-coding sequences in the genome-wide and chloroplast genomic of *Dalbergia* exhibit a preference for A/U-ending codons (Figure 3A and Figure 4). This result is consistent with the previously reported codon usage bias in dicotyledons such as *Theaceae* and *Solanum* [8,48].

The factors that affect codon usage bias are complex, including mutation pressure, GC content, natural selection, gene length, recombination rate, gene expression level, and genetic code repair, among others [49,50,51]. ENC-plot, PR2-plot, and neutrality plot analyses revealed that natural selection is the major factor driving codon usage bias in the genome of *Dalbergia*, with mutation pressure playing a secondary role (Figure 3B–D). This is the same with the previous studies of *Euphorbiaceae* [52], *Panicum* [53], *Malus* [54]. Further exploration is needed into other factors influencing codon usage preferences. Overall, the above-mentioned factors work together to shape the codon usage pattern in *Dalbergia* species. Our analyses suggest that natural selection is the main force driving the observed codon usage bias, which is consistent with the notion that the usage of synonymous codons is under selection for translational efficiency and accuracy. Understanding the factors that govern codon usage patterns in *Dalbergia* could provide insights into the evolutionary processes and molecular mechanisms underlying gene expression regulation in this important genus.

In our study, we evaluated the correlation between the expression levels of protein-coding genes in *D. odorifera* and codon bias in three directions. At the sequence level, we investigated the relationship between gene expression and codon usage bias across high, medium, and low transcription levels. In the high expression gene group, gene expression and codon bias showed a positive correlation, consistent with previous research by *Cuscuta australis* (Convolvulaceae) [34]. Conversely, the medium expression gene group demonstrated a negative correlation between codon bias and gene expression, while the low expression gene group revealed no clear relationship between the two. This may be attributed to the species-specific characteristics of *D.odorifera*. Moreover, the selection of different expression level thresholds in various studies may potentially influence the experimental results. Previous research primarily focused on examining the relationship between gene expression and codon bias in higher expression genes [55]. In our study, we integrated transcription levels for all genes and divided them into three levels to characterize the relationship more comprehensively between gene expression and codon bias in *D.odorifera*. At the codon level, genes with a G/C-ending codon bias showed significantly higher expression levels than those with an A/U-ending codon bias. At the amino acid level, we found that highly expressed genes tended to use G/C-ending synonymous codons. These results are consistent with those observed in *Arabidopsis thaliana* [56]. Overall, the strength of codon bias in gene expression showed a positive correlation, and highly expressed genes tended to use G/C-ending codons.

The standard evolutionary model typically assumes that codon bias is explained by a balance between genetic mutation, selection, and drift (mutation pressure, natural selection, and genetic drift) [57]. We attempted to link the codon usage preference feature with the system evolution feature, seeking to identify similarities and differences between the two. We found that the clustering tree based on the codon usage feature RSCU differed significantly from the other two system evolution trees. The codon usage features of different species in the *Dalbergia* were not entirely identical. Previous literature has reported a negative correlation between codon bias and coding region length, with genes with longer coding regions exhibiting a greater tendency toward random codon usage [58]. The greater the randomness of codon usage is, the less likely the codons are to match together in sequence alignment. Our results confirmed that the clustering based on codon bias and the evolutionary tree constructed from sequence alignment differed significantly (Figure 7 and Appendix A). It is worth noting that the four species, *D. odorifera*, *Dalbergia hainanensis*, *Dalbergia tonkinensis*, and *D. hainanensis*, were more likely to be classified together. We found that these species are geographically close, all located in southern Asia, with *D. odorifera* and *D. hainanensis* originating in China’s Hainan region [19,20,59]. This indicates that species that have a closer origin tend to have more similar codon usage preference features and system evolution features. Additionally, previous research has shown that clustering based on codon bias does not accurately reflect the true system classification and phylogenetic relationships [60]. Our study also confirmed this result. The reasons for the differences may be related to the number of effective codons, the base composition at different positions, and the usage of synonymous codons. This further emphasizes the importance of considering the mutation characteristics of genome sites and sequence information in non-coding regions during the construction of evolutionary trees, thereby helping in the in-depth study of the evolution of *Dalbergia* plants.

## 5. Conclusions

This study investigated the codon usage patterns of the nuclear genome and chloroplast genome of *Dalbergia* species, and explored the correlation between codon usage preferences in the genome of *D. odorifera* and gene expression. Additionally, a phylogenetic analysis of 25 *Dalbergia* species was conducted. The results showed that *Dalbergia* species exhibited a preference for A/U-ending codons at the third position of protein-coding sequences in both the nuclear genome and chloroplast genome. A total of 25 optimal codons was identified in the nuclear genome, and between 15 and 23 optimal codons were identified in the chloroplast genome. Natural selection was found to be the primary factor influencing codon usage bias in *Dalbergia* species, followed by mutation pressure. Furthermore, in *D. odorifera*, genes with stronger codon usage bias exhibit higher expression levels among highly expressed genes, and these highly expressed genes tend to preferentially use G/C-ending codons. The phylogenetic analysis based on chloroplast CDS sequences, and the nuclear genome showed strong similarity, but the phylogenetic tree based on codon usage features (RSCU values) showed significant differences. This suggests that unbiased codons play an important role in the evolutionary process of species. Overall, this study provides insights into the codon usage patterns of *Dalbergia* species and their relationship with gene expression, as well as their evolutionary relationships.

## Figures and Tables

**Figure 1 genes-14-01110-f001:**
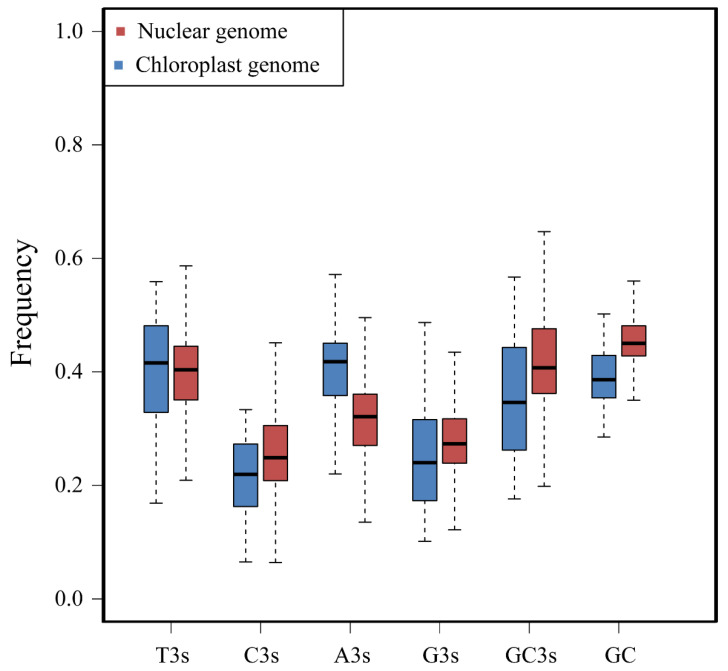
Comparison of base composition between nuclear genome and chloroplast genome.

**Figure 2 genes-14-01110-f002:**
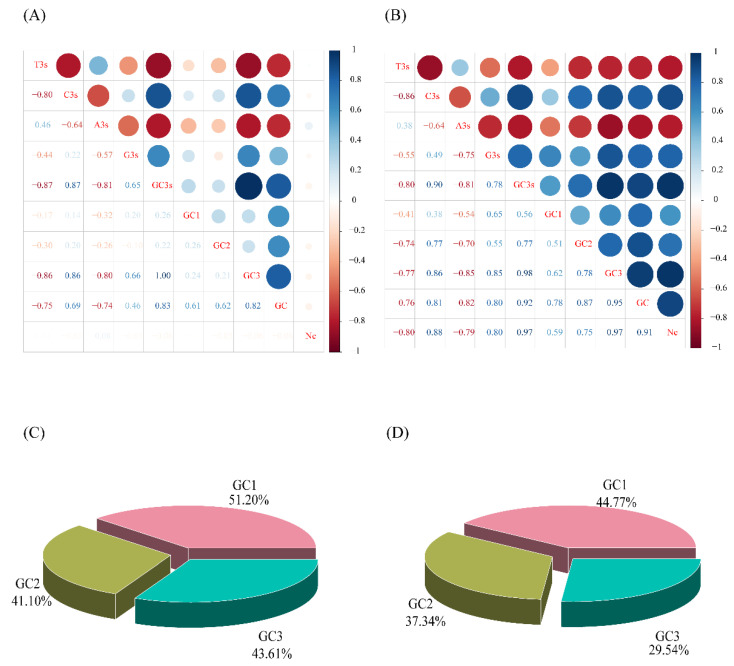
Codon usage parameter characteristics (**A**,**B**) andare correlation of codon usage indicators for protein-coding sequences (The darker the red represents the stronger negative correlation, the darker the blue represents the higher positive correlation. (**C**,**D**) are percentages of GC content at different positions in *D. odorifera* genome. Note: (**A**,**C**) chloroplast genome, (**B**,**D**) Nuclear genome).

**Figure 3 genes-14-01110-f003:**
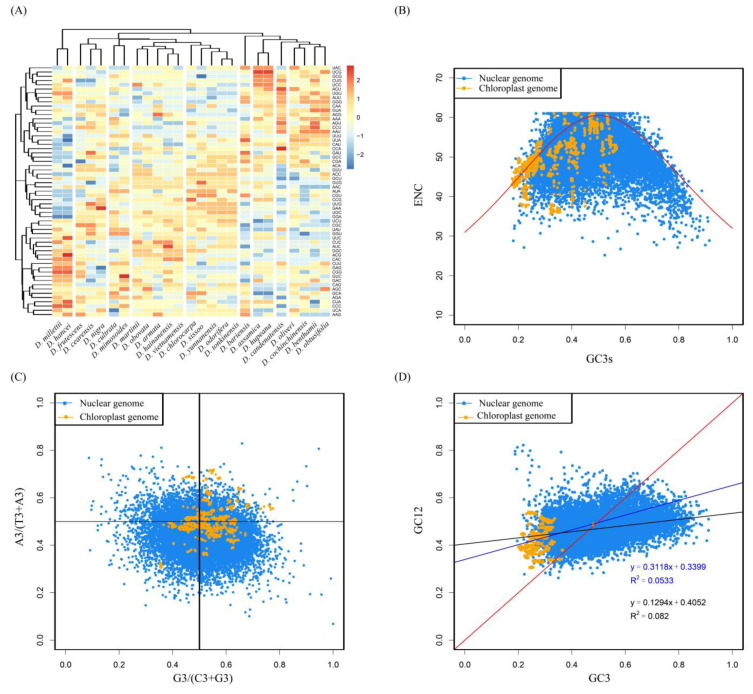
The use of synonymous codons and the factors influencing codon bias (**A**) Heatmap of synonymous codon usage (darker blue shades represent lower RSCU values, while darker red shades represent higher RSCU values). (**B**) ENC-plot analysis of nuclear genome and chloroplast genome. (The red curve is the expected ENC value versus GC3s.) (**C**) Analytical plot of the PR2-plot. (**D**) Neutral plot analysis of nuclear genome and chloroplast genome sequences. (The red diagonal line represents GC3 equals GC12, and if all points are on this line, it suggests that the codon usage pattern is mainly affected by mutations. Otherwise, it is influenced by natural selection.)

**Figure 4 genes-14-01110-f004:**
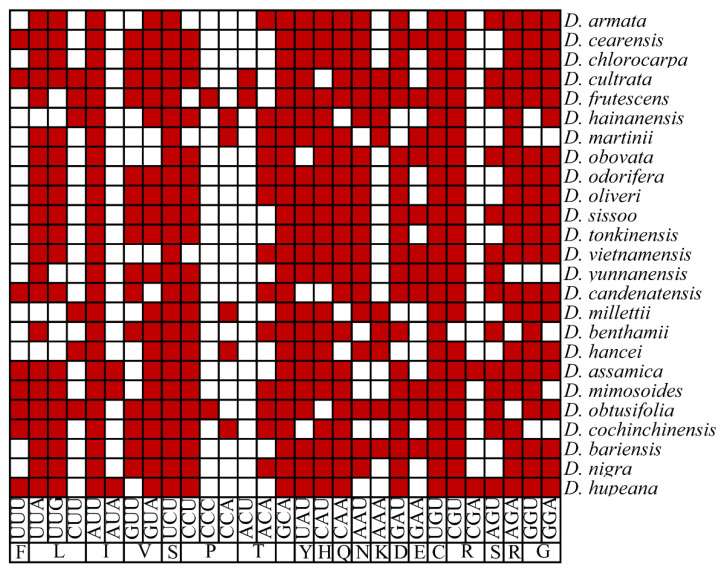
Optimal codon results of 25 *Dalbergia* plants (the optimal codon is indicated in red).

**Figure 5 genes-14-01110-f005:**
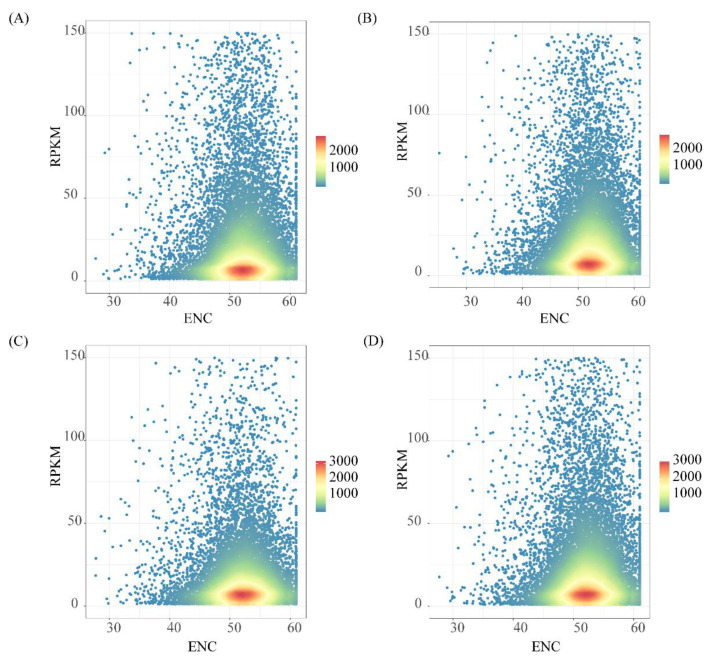
The relationship between gene expression levels and ENC values of different tissues in *D. odorifera* Four-tissue RPKM-ENC distribution heatmap (**A**) leaf, (**B**) root, (**C**) seed, (**D**) stem. The redder the color, the greater the number of neighboring genes, and the bluer the color, the smaller the number of neighboring genes.

**Figure 6 genes-14-01110-f006:**
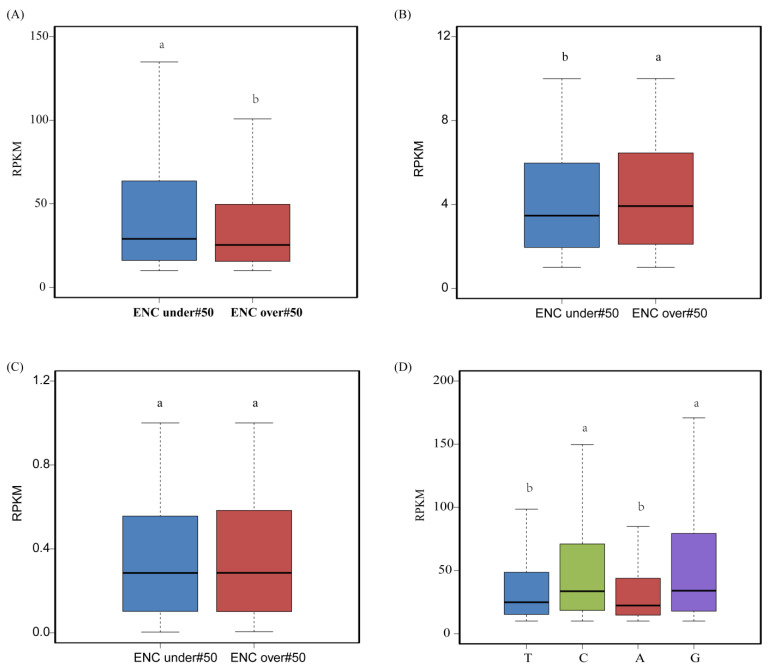
The gene expression of different codon bias in sequence and codon level (**A**) Relationship between high expression gene and ENC value; (**B**) Relationship between medium expression gene and ENC value; (**C**) Relationship between low expression gene and ENC value; (**D**) Relationship between RPKM value and third codon base preference in codon level (Note: In the same boxplot, different letters between groups indicate significant differences (a and b), while identical letters represent non-significant differences (a and a, or b and b).).

**Figure 7 genes-14-01110-f007:**
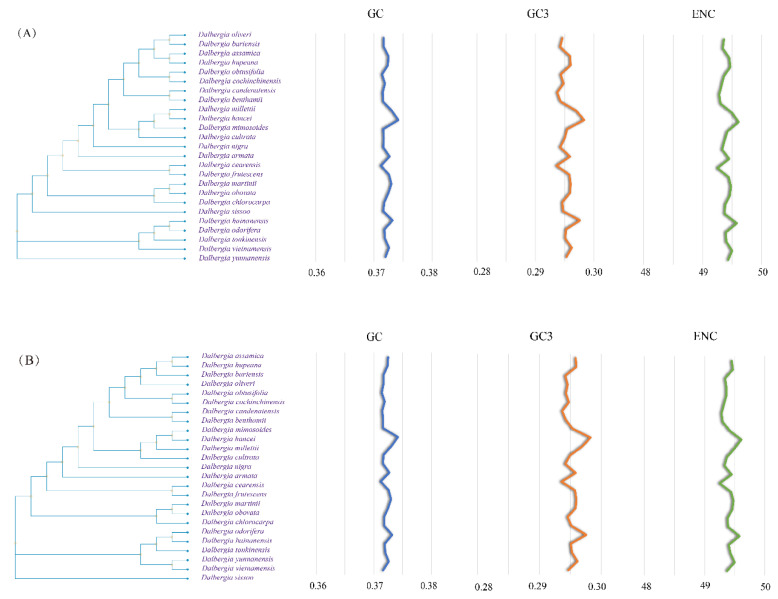
Phylogenetic trees and trend lines of their codon usage indicators (**A**) Phylogenetic tree based on chloroplast protein-coding sequences. (**B**) Phylogenetic tree based on the whole chloroplast genome.

**Table 1 genes-14-01110-t001:** Codon usage indicators in protein-coding sequence of nuclear and chloroplast genome.

Indicators	Nuclear Genome			Chloroplast Genome		
	Mean ± SD	Max.	Min.	Mean ± SD	Max.	Min.
T3s	0.390 ± 0.080	0.6854	0.0317	0.397 ± 0.095	0.5588	0.1684
C3s	0.267 ± 0.087	0.816	0	0.216 ± 0.066	0.3333	0.0649
A3s	0.311 ± 0.073	0.6026	0.0242	0.404 ± 0.076	0.5714	0.2079
G3s	0.283 ± 0.068	0.8381	0.0476	0.253 ± 0.093	0.487	0.1012
ENC	51.86 ± 4.410	61	25.17	49.9 ± 6.199	35.71	61
GC3s	0.432 ± 0.102	0.914	0.175	0.354 ± 0.108	0.567	0.176
GC	0.461 ± 0.050	0.695	0.259	0.390 ± 0.046	0.502	0.285

## Data Availability

The data used in this study are available in the Methods section and Appendix A of the manuscript.

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
