# Peer review of "Comparative Analysis of Codon Usage Patterns in Nuclear and Chloroplast Genome of Dalbergia (Fabaceae)"

_genes, 2023, doi:10.3390/genes14051110_

Round 1

Reviewer 1 Report

In this study, the authors attempted to analyse the codon usage bias of Dalbergia plants using sequence data from gene bank. It's worth noting that genetic code usage preferences can vary among different taxa or genes with different expression levels due to code redundancy. To carry out their analysis, the authors collected coding protein sequences from gene bank. While their study is quite interesting, I have some concerns regarding their comparison strategy. Specifically, I think it would have been beneficial to separate the mRNAs or genes based on high, middle, and low expression levels, as the codon usage bias is known to vary across different expression genes, with housekeeping and inducible genes showing different ratios of codon usage bias. Although the authors used the ENC value to identify high and low expression genomes, it's difficult to discern what sequence data they downloaded and used for analysis. Perhaps it would have been better for comparison if the authors had obtained the transcriptome data themselves.

The english need to polish by native english speaker.

Reviewer 2 Report

I have read the revised manuscript genes-2350472 entitled: Comparative Analysis of Codon Usage Patterns in Nuclear and Chloroplast Genome of Dalbergia for publication in Genes. In this study, authors comprehensively analyzed the CUB patterns of the nuclear genome, chloroplast genome, and gene expression, as well as the systematic evolution of Dalbergia species. This manuscript generally is well-written, and the methodology and conclusions are scientifically sound. I found the paper interesting. The investigations are extensive. The presentation of results is clear and attractive. As such, the manuscript is potentially interesting to a relatively broad readership covering fields like evolutionary biology. My comments mainly relate to relatively minor issues of interpretation and writing. These comments do not influence a positive impression of the article.

Some minor notes: 

Title: Please add the name of the family (Fabaceae) at the end of the titles, should: Comparative Analysis of Codon Usage Patterns in Nuclear and Chloroplast Genome of Dalbergia (Fabaceae)

Line 48-49: Please either remove the abbreviation from the name of the author L. after Ferula or add the abbreviations of their authors to all other genus names. Please consider adding names of botanical families to which the listed genera belong: Achyranthes (Amaranthaceae), Ferula (Apiaceae), and Aquilegia (Ranunculaceae).

Line 51: Dalbergia is a pan-tropical genus with over 269 recognized tree, shrub, and woody vine species [X]. – reference needed

General remark: throughout the manuscript, please adjust the citations of the literature in the text to the rules/recommendations of the MDPI

The text is understandably written, but it can be improved.

Reviewer 3 Report

This is a well written manuscript looking at codon biases in Dalbergia sp., and particularly D. odorifera. The analyses of codon biases are reasonable and are interesting, especially comparison across phylogenies. Not much is said about the reason for this analysis. This work would be useful for future genetic modification of these species, potentially as timber or other economic products. As an N2 fixer this genus can be used as support species for e.g. sandalwood tree production. The economic, or environmental value may be worth mentioning. It may be worthwhile mentioning this, to give some background to their interest and motivation for the work. If the authors are purely interested in the scientific study of chloroplast and nuclear codon biases, then some comparison to other plant genera might add some more interest and readership and give some motivation to the work. Any of these options increases the background story and would add interest in the paper.

Round 2

Reviewer 1 Report

the authors already revised their manuscript based on the reviewer's comments and the manuscript is on good condition for next step.